# The Changing Patterns and Correlates of Adolescent Substance Use in China’s Special Administrative Region of Macau

**DOI:** 10.3390/ijerph19137988

**Published:** 2022-06-29

**Authors:** Spencer De Li, Lu Xie, Kehui Wu, Jiaqi Lu, Mi Kang, Hui Shen

**Affiliations:** Department of Sociology, University of Macau, Macau 999078, China; spencerli@um.edu.mo (S.D.L.); yc17311@connect.um.edu.mo (L.X.); mc15181@connect.um.edu.mo (K.W.); mikang90@outlook.com (M.K.); yc17328@connect.um.edu.mo (H.S.)

**Keywords:** cigarette, alcohol, illicit drug, adolescent, substance use, Macau

## Abstract

Most of the research on adolescent substance use is from either the U.S, Europe, or other non-Eastern countries, but very little attention is paid to that in the Greater China Region. As a special administrative region of China, Macau is known for its gambling industry, its proximity to the Golden Triangle, and its lenient drug laws, all of which can be conducive to high-level drug use in the population, including its adolescents. Yet, the extent and patterns of adolescent substance use in Macau are not well understood. Using the data collected from two large representative samples of secondary school students in 2014 and 2018, this study provided population-based estimates of the prevalence rates of lifetime and past 30-day substance use among Macau adolescents in the two separate survey years. By comparing the two sets of estimates, it established the patterns of changes in cigarette smoking, alcohol use, and illicit drug use among adolescents in Macau during the period. Additionally, through the analysis of the data in the latest year, it identified risk factors for adolescent substance use in the special administrative region. Among the key results, the rates of cigarette smoking and illicit drug use were low to moderate while the rates of alcohol use were relatively high; cigarette smoking decreased during the period, but alcohol use and illicit drug use remained stable; Macau adolescents shared similar risk factors for substance use with adolescents elsewhere, but unique circumstances might exist to influence their alcohol consumption.

## 1. Introduction

Substance use, including the use of tobacco, alcohol, and illicit drugs has been found to be associated with many health and social problems throughout the world [1,2,3,4,5]. Adolescents are always more vulnerable to taking risks and engaging in unhealthy or delinquent behaviors, such as cigarette smoking, alcohol drinking, and using illicit drugs [6]. Previous studies have shown that many adolescents have tried alcohol, cigarettes, and other drugs at an early age [7,8]. In the U.S., at least 78.2% of adolescents had tasted alcohol, about 45% had tried cigarettes, and roughly 14–20% had tried other stimulants and hallucinogens by late adolescence [9,10,11,12]. Administered in about 36 countries, the European Survey Project on Alcohol and Drugs (ESPAD) collects information on drug use by adolescents in Europe. The most recent ESPAD, in 2019, reported that an average of 18% of students age 13 or younger had smoked cigarettes, 33% had tried alcohol, 6.7% had experienced alcohol intoxication, and 2.4% had used marijuana [13]. Early onsets of substance misuse by adolescents were not only associated with various substance-related problems [7], but also increases their risks of adulthood substance dependency [14,15,16,17,18,19] and other undesirable outcomes (e.g., sexually transmitted disease infection, early pregnancy, and crime) [20]. Therefore, it is critical to identify and prevent early substance use among adolescents.

In the Greater China Region (GCR) of Mainland China, Hong Kong, Macau, and Taiwan, population-based surveys on the prevalence and trend of adolescent substance use are scarce, although some data are available on certain types of substance use among selected groups of adolescents or young adults. For example, a national survey of middle school students conducted by the Chinese Center for Disease Control and Prevention showed that, between 2014 and 2019, the lifetime and current use of cigarettes decreased moderately, while the current use of e-cigarettes increased significantly [21]. For illicit drug use, according to the official reports issued by the National Narcotics Control Commission of China, the number of registered adolescent users decreased tremendously. In 2014, there were 29,000 drug users under the age of 18 (accounting for 0.98% of all drug users) [22], and by 2019, there were only 7151 drug users under the age of 18 (accounting for 0.3%) [23].

Macau is a special administrative region, situated in southern China, with its own political, economic, and legal systems. It is close to the Golden Triangle, one of the biggest illicit drugs producing regions in the world, that borders mainland China, which has rapidly become a major source of synthetic drugs, especially methamphetamine, ketamine, and fentanyl. Macau is also known as the “Monte Carlo of the Orient” and has surpassed Las Vegas by hosting the world’s largest gambling industry in terms of revenue and profit [24]. Additionally, Macau has very lenient drug enforcement laws with a maximum length of imprisonment of only 12 months for drug use and 15 years for drug trafficking [25]. Much of the knowledge on the trends in substance use in Macau is based on official data provided by relevant government agencies, which tend to underestimate the true extent of drug use. This problem is especially acute nowadays considering that the popularity of new psychoactive substances has led to widespread hidden drug use among adolescents [26].

Overall, evidence indicates that adolescents from different parts of the world use both licit and illicit substances at different rates and patterns. There are persisting uncertainties about the variations in the prevalence and trend of adolescent drug use, with population heterogeneity, normative system, peer association, social environment, and perceived availability of drugs being potential risk factors. Studies conducted in the West have supported the influences of these factors on adolescent drug use [27,28,29,30]. However, there is still a lack of empirical research on the trends and predictors of adolescent drug use in the GCR. As a result, it remains unclear if the trends observed in other parts of the world can describe adolescent substance use in the GCR region. Likewise, the predictors of adolescent substance use observed in the West may not have the same effects in the GCR. Henceforth, culturally specific studies are essential, and this study tries to contribute to past research by examining the trend and risk factors associated with adolescent drug use in Macau. Through the collection and analysis of the survey data from large representative samples of secondary students in the region in 2014 and 2018, this study aims to estimate the prevalence rates of adolescent drug use, identify the patterns of change in behavior during the period, and examine the relationships between risk factors and substance use among adolescents in Macau.

## 2. Literature Review

The topic of adolescent substance use is of interest because the early exposure to licit and illicit drugs increases as children transition into adolescence [31]. The health and social consequences of licit and illicit drug use have led countries to continually acquire statistical data that keeps track of substance use by adolescents. Drug use reports and drug-related research across the world have provided important insights into the trends and statistics for adolescent substance use. The knowledge gained from these empirical research and trend reports is critical for the development of evidence-based policy to prevent and control the problem of adolescent drug use.

According to the Global School-Based Student Health Survey (GSHS) in 57 low- and middle-income countries, between 2008 and 2013, there were about 25% of 12- to 15-year-old adolescents who have tried at least one alcoholic drink in the past month and 17.9% reported that they had become drunk at least once in their lifetime [7]. For the past two decades, studies conducted in the U.S., Europe, and other high-income countries revealed a downward trend in the prevalence of cigarette smoking and alcohol use among adolescents, yet trends in illicit drug use were heterogeneous [13,30,32]. However, data from selected Asian countries indicated a much lower prevalence in smoking and illicit drug using than those reported in the West [33]. Thus, substance use by adolescents of licit and illicit drugs is a worldwide problem with strong contextual variations. To fully grasp the scope and the nature of the problem, we need to understand not only worldwide trends but also the local patterns of variations and the circumstances that shape these changes.

### 2.1. Trends in Adolescent Substance Use

#### 2.1.1. Alcohol Use

According to the Monitoring the Future (MTF) surveys of U.S. adolescents in grades 8, 10, and 12 from 2000 to 2019, the prevalence of lifetime and past 30-day alcohol use declined sharply [32]. There were sizable decreases in the past 30-day use of alcohol between 2010 and 2015, with 13.8%, 28.9%, and 41.2% for 8th, 10th, and 12th graders in 2010 down to 9.7%, 21.5%, and 35.3% in 2015, respectively [34]. Similar patterns were also observed in other Western countries. Reports from ESPAD indicates a downward trend in the current use of alcohol in Europe, with estimates to be about 57% in 2011, 48% in 2015, and 47% in 2019 for adolescents in ages 11 to 15, with the frequency of 15 years old pupils trying alcohol being seven times that of 11 years old pupils [13,35,36].

In Hong Kong, the surveys conducted by the Narcotics Division of Security Bureau showed that, from 2011 to 2018, the proportions of lifetime alcohol-taking in students slightly increased from 56.0% to 56.7% [37]. In Taiwan, data from the 2017 National Health Survey showed that the prevalence of alcohol among adolescents aged 12–17 slightly decreased from 25.7% to 23.8% in 2009 through 2013 and then increased up to 27.7% in 2017 [38]. Survey data collected from fifth graders through college seniors sponsored by the Social Welfare Bureau of Macau SAR showed a decreased prevalence in alcohol use of 63.16%, 56.14%, and 52.48% in 2010, 2014, and 2018, respectively [39,40,41].

#### 2.1.2. Cigarette Use

The MTF surveys conducted between 2016 and 2020 showed a slight decrease in the lifetime use of cigarettes by adolescents in the U.S. Furthermore, for the same period, the prevalence of past 30-day use of cigarettes by adolescents from grades 8, 10, and 12 in the U.S. decreased from 2.6% to 2.2%, 4.9% to 3.2%, and 10.5% to 7.5%, respectively [42]. According to the ESPAD, the prevalence of current adolescent smoking in Europe is slightly higher than that in the U.S. Between 2011 and 2019, the prevalence of adolescents who were considered current smokers (smoked in the past 30 days) in Europe decreased by 10%, but this figure only accounted for adolescents between the ages of 11–15 [13]. Thus, it is likely that there would be a higher prevalence figure since the trends omitted adolescents who were older than 15 years old.

In Mainland China, according to the data collected from a series of cross-sectional National Health Service surveys conducted between the years of 2003 and 2013, smoking prevalence among adolescents aged 15–24 years increased from 8.3% in 2003 to 12.5% in 2013 [43], and the figure climbed to 18.6% in 2018 [44]. In Hong Kong, the Narcotics Division of Security Bureau found that, among all surveyed students, the proportions of lifetime tobacco-taking students somewhat decreased from 8.8% in 2011 to 7.0% in 2018 [37,45]. The Global Youth Tobacco Survey in Taiwan reported that the prevalence of smoking cigarettes among middle school students decreased from 8.0% in 2010 to 3.0% in 2019, while the prevalence among high school students decreased from 14.8% in 2009 to 8.4% in 2019 [46]. In Macau, government reports showed that the prevalence rates of smoking cigarettes among adolescents were 11.95%, 14.16%, and 10.84% in 2010, 2014, and 2018 [39,40,41].

#### 2.1.3. Illicit Drug Use

In the U.S., the lifetime use of any illicit drug in adolescents declined gradually between 2000 and 2010 and was unvarying from 2019 and onwards; while the past 30-day use slightly decreased from 2000 to 2010 and remained stable starting from 2019 [32,34]. Marijuana is one of the many illicit substances that adolescents in Western countries frequently use because it is easily accessible and growing in popularity [11]. The past 30-day use of marijuana rose from 5.8% for 8th graders, 13.8% for 10th, and 19.4% for 12th graders in 2008, to 7%, 18%, and 22.7% in 2013 for the same grade group, respectively [47,48]. In Europe, between 1995 and 2011, the lifetime use of illicit drugs, like cocaine, ecstasy, amphetamine, and LSD, increased from 12% to 20% but started to decrease slowly after 2011 [13]. About 16% of adolescents have tried marijuana during some time in their life in the whole of Europe. The current (past-30-day) use of marijuana increased the greatest between 1995 and 1999, but the increase eased beginning in 2003 [13]. The highest prevalence of marijuana use that occurred in 2011 was 7.6% but remained relatively stable in the years after that [13]. Thus, for the past decade, the trend in illicit drug use in both the U.S and Europe displayed a downward trend in both lifetime and current use of illicit drugs among adolescents.

As discussed earlier, official annual reports issued by the National Narcotics Control Commission of China indicated that the number of registered drug users, including adolescent users, decreased dramatically between 2014 and 2019 [22,23]. According to the data collected by provincial governments in China in 2016, 8% of all registered drug users were individuals aged 25 and below [49]. Because the reporting agencies did not treat adolescents as a separate category, it is not clear how many of the young drug users were adolescents. Official statistics from Hong Kong also showed a large decrease in the number of people under the age of 21 who used illicit drugs, with the proportion of total drug users dropping from 17% in 2011 to 9% in 2020 [50]. The large decrease shown in the official statistics caused some scholars to question the reliability of the data that might have been undermined by “hidden drug use” [51] (p. 10). In Taiwan, statistics reported by the Ministry of Health and Welfare indicate that illegal drug use among adolescents decreased gradually between 2012 and 2020 after prior increases between 2007 and 2012 [52,53]. In Macau, reports from the Social Welfare Bureau showed that the prevalence of drug use among students from fifth graders to college undergraduates was 1.88%, 2.48%, and 2.92% in 2010, 2014, and 2018, respectively [39,40,41]. The data indicate a general increasing trend although the change from 2014 to 2018 was not statistically significant.

### 2.2. Risk Factors for Adolescents’ Substance Use

Considerable research has shown that adolescents are more likely to use psychoactive substances under the influences of certain risk factors. Family structure, parental supervision, peer association, school problems, the availability of drugs, and normative attitudes toward drug use have been frequently identified as predictors of cigarette, alcohol, and illicit drug use among adolescents [54,55].

Family structure can influence alcohol and illicit drug use in adolescents as growing up in single-parent households tends to increase the likelihood of using alcohol and drugs [56]. Relative to intact families, children growing up in single-parent families receive less parental monitoring, and inadequate parental monitoring makes the children more likely to engage in alcohol and other substance use as adolescents [57]. Likewise, adolescents residing in stepparent families are at an increased risk of becoming substance users, because these families tend to be more fragmented in terms of the bonding between the children and the stepparents [58].

Good family functioning, characterized by parental supervision and parental support, is also found to significantly impact the level of substance use in adolescents [59,60,61]. In most urban areas, like Macau, consistent involvement and support from parents in their children’s daily activities is challenging, because most of the parents have nonstandard work arrangements (e.g., shift work). Parents working with nonstandard schedules often have difficulties providing adequate parental involvement and monitoring for their children [62]. In addition, the nonstandard working schedule of parents usually impacts parent-child interaction and connectedness, which affects children’s developmental trajectory subsequently.

In addition to parents, peers can also play an important role in influencing adolescents’ substance use. Adolescence is characterized by a stage of autonomy development, and most of the connections established by adolescents during this period will be orientated towards their peers and less to their parents. Peer acceptance is a major component in adolescents’ development of autonomy because it is associated with high self-esteem and social competence [63,64]. Thus, to gain acceptance from peers, adolescents tend to mimic the behavior of their peers to acquire the identity of belonging to the peer group.

Furthermore, much research in the U.S. and Europe shows that lower school engagement or absenteeism can increase the risk of substance use in adolescents. Problematic school behaviors in adolescents are related to substance use including school absenteeism or truancy and poor academic performance [65]. School absenteeism or truancy is a robust precursor to subsequent drug use because truant youths, or absentees, are more likely to be exposed to licit and illicit substances than adolescents who are more committed to school [66,67]. The probability of smoking tends to be higher for students who have the least school engagement and face the greatest schoolwork difficulties [68]. Conversely, greater engagement in school was associated with less precocious substance use behavior, such as drinking, in most adolescents.

The availability of drugs can also systematically influence adolescents’ substance use by increasing their opportunities to obtain and use those substances. Prior research has indicated that when adolescents have easy access to alcohol, they are 3.5 times more likely to use alcohol and drink hazardously [69]. In addition, research that examined the relationship between the perceived availability of nicotine and the frequency of nicotine vaping revealed a connection between them [70].

In addition to the risk factors reviewed above, belief also plays an important role in substance use. Lower disapproval and perceived harm of drugs are correlated with a higher level of substance use [27,71,72]. Furthermore, the use of legal substances (i.e., alcohol, tobacco, and betel nuts) can increase the likelihood of illicit drug use [73,74,75,76,77]. Finally, demographic and socioeconomic variables, including sex [74,78,79], age [75,77], place of residence [78], educational levels [78], and income [80], have been reported as related to substance use.

However, most of the studies reviewed above were conducted outside the GCR. There is a lack of research on the trends in substance use among adolescents in the GCR other than government reports based on official data, such as the China Drug Situation Report, Annual Report of National Drug Abuse Monitoring, and the Central Registry of Drug Abuse Information of the Narcotics Department in Hong Kong. The survey results cited above concerning adolescent substance use in Macau were also based on unpublished reports released by the local government that are available only in Chinese. Government reports typically concentrate on the descriptions of frequencies and rates of illicit drug use, with limited attention to theoretically based risk and protective factors for adolescent substance use. Research using representative samples of adolescents in the GCR is needed to gain a better understanding the levels of adolescent drug use, its trend, and its risk and protective factors. Another issue that this study addresses is the inconsistency in the definition of adolescent in the studies conducted in the GCR, which have included samples of disparate age groups. For instance, the surveys of Student and Drugs in Macau covered youth between the ages of 10 and 22, whereas the Survey of Street Youth Drug Abuse included youths between 12 and 24. Comparison of adolescent drug use trends between the GCR and those of the other countries are difficult when different youth age groups are used. To address this problem, the current study collected data from secondary students aged 10–23. Because of the grade retention system widely implemented in Macau secondary schools, some students remained in school until their early 20s [81]. Students who were younger than 11 and older than 20, however, made up only about 0.2 percent of the samples in 2014 and 2018. The rest of the sample in either survey year consisted of students aged 11–20, an age group that is more consistent with the definition of adolescents adopted in studies conducted in other countries, especially in Europe and the U.S.

## 3. Materials and Methods

### 3.1. Data

The data in the present study were collected from two cross-sectional surveys on middle school and high school students in 2014 and 2018. The surveys were part of the multiyear project, titled “Student and Drugs in Macau (SDM),” which was funded by the Macau Social Welfare Bureau, the government agency overseeing all drug use prevention and treatment programs in Macau. To increase the representativeness of the sample, both the 2014 and 2018 surveys used a multi-stage stratified cluster probability proportional to the size sampling method. The first stage of sampling involved dividing all middle schools and high schools into seven strata based on their located district, and sampled schools were selected from each stratum using a probability-proportional-to-size sampling procedure. In the second stage, we randomly selected one to two classes from all eligible grades (i.e., seventh, eighth, ninth, tenth, eleventh, and twelfth grades) at each sampled school. All students in the selected classes were invited to participate in the surveys. Those who agreed to participate signed an informed consent form and were provided with a questionnaire that was designed using Remark OMR software. Trained research staff appeared onsite to distribute and collect the questionnaires and to answer students’ questions. The responses to the survey were directly scanned into a data table and manually verified by the research staff. We obtained 6801 and 6464 completed questionnaires in 2014 and 2018, respectively, which represented 98.5% (of 6904) and 99.9% (of 6467) of the sampled students in the two separate years.

### 3.2. Measures

**Substance use.** Questions on substance use were based on instruments used in Monitoring the Future (MTF) [82], and we modified some of the questions to make them more relevant to the local context of Macau. Specifically, we excluded drugs that adolescents in Macau may have never or rarely reported using and added the names of drugs that were commonly observed but were not on the MTF instrument. We examined respondents’ cigarette, alcohol, and illicit substance use (0 = no, 1 = yes) in two different time spans: lifetime use (Cronbach’s α = 0.78) and past 30-day use (Cronbach’s α = 0.79). The past 30-day use was considered as current use and was analyzed based on grade-by-gender time trends and was used as a dependent variable in our multivariate analyses. Illicit drug use was identified as any use of the following nominated substances: ketamine, ecstasy, crystal meth, heroin, marijuana, cocaine, codeine, pills (mostly new psychoactive substances such as valium), happy water (mixed drugs prevalent in Macao), etc. In addition, participants were asked, “How many times (if any) have you been drunk in the past 12 months?”. Based on their responses to this question, we created a variable measuring drunk experiences in the past 12 months with “0 = no” and “1 = yes”.

**School problems.** Participants were asked about their truancy experience (“for the past 30 days, how many times have you missed school because of truancy?”); dropout experience (“have you ever dropped out of school?”); school punishment experience (“have you ever been punished because of misbehaviors at school?”). Participants with any truancy, dropout, or school punishment experience were identified as having school problems. School problems were coded with “0 = no” and “1 = yes”.

**Awareness and attitude toward substance use**. We used two variables to measure participants’ attitudes toward substance use: (1) perception of risk was measured by the question: “How much risk will people experience if they are involved in the following behaviors?”. The behaviors were smoking cigarettes, drinking alcohol, and using illicit substances (the nine nominated substances mentioned above) (Cronbach’s α = 0.93). Responses for the perceived risk were coded as “0 = unknown/none/slight/moderate risk, 1 = great risk”; (2) approval of substance use was examined by the question: “How much do you approve of people engaging in the following behaviors?”, addressing the same group of substances (Cronbach’s α = 0.95). Responses for the approval of substance use behavior were coded with “0 = unknown/somewhat/strongly disapprove” and “1 = somewhat approve/strongly approve”. The purpose of selecting these two variables is to examine participants’ awareness and attitudes toward cigarette smoking, alcohol use, and illicit drug use.

**Availability of drugs.** We asked the adolescent respondents “How difficult would it be for you to obtain the following substance if you wanted some?”. The items included the nominated substances mentioned above (Cronbach’s α = 0.99). Responses for this variable were coded with “0 = fairly/very difficult or probably impossible” and “1 = fairly/very easy for any nominated substances”.

**Peer substance use.** We asked the respondents “How many of your friends use the following substance(s)?”. Based on the adolescents’ responses, we counted the number of their peers who smoked cigarettes, drank alcohol, and used illicit drugs (the nominated substances mentioned above) (Cronbach’s α = 0.82). Responses to peer substance use were coded with “0 = none” and “1 = at least one”.

**Family characteristics.** Participants were asked about their family structure (whether both parents are alive, only father is alive, only mother is alive, or both parents are deceased) and parental employment status (whether both parents are employed, just father is employed, just mother is employed, both parents are unemployed). We used parental employment as a proxy for parental supervision. Because gambling and tourism are the dominating industries in Macau, shift work is commonly unavoidable for workers in either of the industries. Hence, children become unsupervised if their parents, or parent of children from a single-parent family, work in those industrial sectors. As a result, adolescents’ use of drugs at home can escalate because of the lack of parental supervision due to their parents’ nonstandard work schedules.

**Sociodemographic variables**. The student’s gender, age, grade, birthplace (Macau, mainland China, Hong Kong, others), and monthly pocket money (0 MOP, 1–500 MOP, 501–1000 MOP, more than 1000 MOP) were included in our regression analyses as control variables.

### 3.3. Statistical Analysis

We used data from the SDM surveys in 2014 and 2018 for our analyses. We first described the sample characteristics and the prevalence of lifetime and past 30-day use of cigarette, alcohol, and illicit substance in 2014 and 2018 (Table 1). Next, bivariate analyses were conducted to examine differences in substance use associated with demographic factors. Then we compared the rates of substance use prevalence among grade-by-gender groups for both 2014 and 2018. After that, we conducted multivariate logistic regression analyses using data from 2018 to identify factors associated with past 30-day use of cigarettes, alcohol, and illicit substances. Any missing data were excluded from regression analyses, and all the included data were analyzed using Stata 17.0.

## 4. Results

### 4.1. Sample Characteristics

After excluding the missing values, we analyzed a total sample of 6782 students in 2014 and 6459 students in 2018, both of which included students from Grade 7 through Grade 12 in Macau. Additionally, the 2014 sample was made up of 46.9% female and the 2018 sample was made up of 46.0% female. Across the two SDM surveys, the respondents’ profiles remained stable in terms of their gender and birthplace, while the proportion of upper-grade students in the sample was lower in 2018 (Table 1). The mean age of the samples from both 2014 and 2018 was 15 years old (SD: 2.0). The demographic characteristics, attitude toward substance use, and perceived access to illicit drugs in both samples are shown in Table 1. Table 2 presents more in-depth information on the 2018 sample, including respondents’ monthly pocket money, school problems, drunk experience, family characteristics, and peer substance use, all of which were included in the multivariate logistic regression analysis. Overall, 1.3% of the participants with missing values in the 2018 survey were excluded from the multivariate regression analyses. Details on sampling, response rates, and missing data can be found in the Appendix A (see Table A1).

### 4.2. Trends in the Prevalence of Cigarette, Alcohol, and Illicit Substance Use

In each survey, respondents who used cigarettes were more likely to be males, Hong Kong-born, foreign-born, or high school senior students (grade 11 and 12). Respondents who used alcohol were more likely to be high school senior students, while those who used illicit drugs were more likely to be males and foreign-born adolescents. The differences in alcohol use by gender and birthplace were only significant in 2014 (refer to Table 1).

Figure 1 displays the changes in the prevalence rates of lifetime substance use of cigarettes, alcohol, and illicit drugs between 2014 and 2018. The prevalence rate of cigarette smoking significantly declined from 13.9% in 2014 to 9.7% in 2018 (*p* < 0.001), while there were no significant changes in the prevalence of alcohol and illicit drug usage for the same years.

Among the 10 nominated illicit substances mentioned above, the prevalence rate between 2014 and 2018 declined from 3.1% to 2.3% for codeine (*p* < 0.01) but increased from 0.8% to 1.2% for crystal meth (*p* < 0.05), 0.6% to 1.4% for pills (*p* < 0.001), 0.6% to 1.1% for heroin (*p* < 0.01), 0.6% to 1.1% for ecstasy (*p* < 0.001), 0.5% to 1.4% for happy water (*p* < 0.001), and 0.6% to 2.1% for others (*p* < 0.001) (see Figure 2).

Figure 3 illustrates the grade-by-gender prevalence rates of past 30-day use of cigarettes, past 30-day use of alcohol, past 30-day use of illicit drugs, and past 12-month drunk experience. The prevalence of past 30-day use of cigarettes declined in almost all subgroups between 2014 and 2018. The most notable decrease was observed in girls who were in grade 11, revealing a decrease from 6.6% in 2014 to 2.3% in 2018. However, the prevalence of alcohol use over the past 30-day use increased in senior boys (grades 11 and 12) and almost all girls in grades 8 to 11. Although a general increase in the use of alcohol was observed among female students (from 23.2% in 2014 to 25.3% in 2018), the prevalence rate of past 12-month drunk experience among female students declined from 9.9% in 2014 to 7.6% in 2018. There was also an increase in the prevalence of past 30-day use of illicit substances among senior boys (grade 11 and 12); conversely, there was a decrease amongst senior girls (see Figure 3).

### 4.3. Factors Associated with Past 30-Day Use of Cigarette, Alcohol, and Illicit Substance

Table 3 shows results from multivariate logistic analyses using data from the 2018 survey (*n* = 6459). After adjusting for gender and grade, we found that, compared to adolescent respondents born in Macau, foreign-born respondents were more likely to use cigarettes (OR = 2.92, *p* < 0.01) and those born in Mainland China were less likely to use alcohol (OR = 0.78, *p* < 0.01). Monthly pocket money was negatively related to cigarette use and positively related to alcohol use. Compared to respondents who had 0 MOP pocket money per month, respondents with monthly pocket money of 1–500 MOP and 500–1000 MOP were less likely to use cigarettes (OR = 0.30, *p* < 0.001 and OR = 0.46, *p* < 0.05), but those with monthly pocket money equal to or more than 1000 MOP were more likely to use alcohol (OR = 1.26, *p* < 0.05; OR = 1.42, *p* < 0.01; OR = 1.77, *p* < 0.001).

Compared to respondents whose parents are both alive, those whose parents are both deceased were more likely to use cigarettes (OR = 8.27, *p* < 0.01). With regards to parental employment, respondents with both parents unemployed were more likely to use cigarettes, in contrast to those whose parents were both employed. (OR = 2.13, *p* < 0.05).

Attitudes toward substance use significantly predicted the respondent’s substance use behaviors. Greater approval of substance use was positively related to respondents’ cigarette use (OR = 10.27, *p* < 0.001), alcohol use (OR = 2.87, *p* < 0.001), and illicit substance use (OR = 3.54, *p* < 0.001). Additionally, a higher perceived risk of substance use was negatively related to the adolescent’s use of cigarettes (OR = 0.35, *p* < 0.001), alcohol (OR = 0.79, *p* < 0.001), and illicit drugs (OR = 0.43, *p* < 0.001). Moreover, the perceived availability of drugs was positively related to illicit drug use (OR = 2.16, *p* < 0.001).

The influence of peers’ substance uses on the respondent’s substance use behaviors displayed various patterns. Peers’ cigarette use was positively related to the respondent’s cigarette use (OR = 3.35, *p* < 0.01), and peers’ alcohol use was positively related to the respondent’s alcohol use (OR = 1.53, *p* < 0.001) but negatively related to the respondent’s cigarette use (OR = 0.28, *p* < 0.01). Also, peers’ illicit drug use was positively related to the respondent’s illicit drug use (OR = 6.27, *p* < 0.001).

The results also show that the respondent’s use of cigarettes and alcohol use were intertwined with each other. Respondents with more alcohol use and drunkenness experience were more likely to smoke cigarettes (OR = 2.89, *p* < 0.001; OR = 5.42, *p* < 0.001, separately), and those who smoked cigarettes and had got drunk before were much more likely to be a current alcohol user (OR = 2.40, *p* < 0.001; OR = 8.72, *p* < 0.001, separately). However, only cigarette use was positively related to illicit drug use (OR = 5.60, *p* < 0.001). Lastly, we also found that school problems were positively related to both cigarette smoking and alcohol use (OR = 1.91, *p* < 0.01; OR = 1.29, *p* < 0.05, separately).

## 5. Discussion

Across Europe and U.S., there has been a general declining trend in cigarette smoking and alcohol use among adolescents in recent years, while the rates of illicit drug use have remained relatively more stable. In the Greater China Region, which comprises Mainland China, Hong Kong, Taiwan, and Macau, adolescent substance use has shown diverse prevalence rates and mixed patterns of change. To a large extent, the inconclusive findings are due to the lack of population-based surveys in the region. Data from official reports and non-representative surveys tend to provide mixed results concerning the extent of and trend in adolescent substance use.

To generate a more reliable estimate of the rates of adolescent substance use, we collected data on substance use and its correlates from two large representative samples of middle and high school students in Macau between 2014 and 2018. Our results showed that, compared to their counterparts in Europe and U.S., Macau adolescents reported low-to-moderate levels of cigarette and illicit drug use but a high level of alcohol use. Specifically, 9.7% of the adolescent respondents reported tobacco use while 4.6% of them reported having used illicit drugs in their lifetime during 2018. The prevalence of past 30-day use of cigarettes and illicit drugs were 3.1% and 1.9%, respectively. In contrast, 53.2% of the adolescent reported alcohol use in their lifetime while 26.0% reported having used alcohol in the past 30 days.

The difference in the levels of consumption between cigarette and illicit drug use on the one hand and alcohol use on the other is not surprising. In recent years, Macau, in general, has significantly stepped up its effort to prevent adolescent use of tobacco and illicit drugs. Alcohol use, however, is more socially acceptable in the region. Macau has no regulations that restrict the purchase and consumption of alcohol by minors, so even children of elementary school age or younger can buy or drink alcohol without prohibition applied to them. The greater latitude toward alcohol has led to higher levels of drinking behavior in most age groups in the population, including adolescents.

The changes in the rates of last 30-day substance use among the Macau adolescents from 2014 to 2018 bore some similarities with the trends in Europe and U.S., but they maintained their unique patterns. Like Europe and the U.S., the rate of cigarette smoking decreased while the rate of illicit drug use remained relatively stable. Unlike Western societies, however, Macau’s adolescent alcohol use did not show any changes during the specified periods. Again, the stability in the rates of alcohol use among Macau adolescents is likely attributable to the lack of restriction on adolescent alcohol use and the society’s acceptance of unrestricted drinking behavior among different age groups.

Among the individual drugs covered in the surveys, the substances that bore the street names of “pills”, “happy water”, and those in the “others” categories showed the largest increases from 2014 to 2018. These findings are consistent with the overall trends of substance use in the GCR in the last few years. Many government reports and self-report surveys have indicated that the use of new psychoactive substances (NPS) is on the rise among adolescents in the region [83]. The categories of drugs with the largest increases encompass some form of new psychoactive substance, which to a large extent accounted for their increase in popularity. Because many of these drugs are comparatively new, their harms have not been fully studied. Thus, the popularity and influences of these drugs among adolescents should be monitored more closely.

The grade-by-gender analysis shows that the rate of past 30-day cigarette smoking in 2018 were consistently lower than that in 2014 for all grade and gender groups. The differences between the two survey years in past 30-day alcohol use, illicit drug use, and getting drunk were mixed and mostly moderate across all grade-by-gender groups, except for past 30-day illicit drug use in the female adolescent groups in grades 9 to 11 and male adolescent groups in grades 11 and 12. Overall, the upper-grade male adolescent group, especially those in grade 12, was the group with the highest risks. They had the highest rates in past 30-day alcohol use, illicit drug use, and getting drunk. The rates of substance use in these three areas were the highest in their respective categories in both years. These findings show that, in terms of both the level and rate of change, male adolescents in the 12th grade had the highest risk for substance use. Hence, this grade group should become a focus of drug use prevention programs that target at-risk adolescents and youth.

Regarding risk factors for adolescent substance use, our analysis identified several factors that had been cited in studies conducted elsewhere, especially in Europe and the U.S. Specifically, we found that living without a parent, parental unemployment, the amount of pocket money, approval of substance use, low perception of risk from substance use, peer substance use, school problems, and the availability of drugs all predicted at least one form of substance use among the adolescents. Furthermore, cigarette smoking was positively related to illicit drug use. There were also two findings related to alcohol use that are not commonly seen in previous research. First, the amount of pocket money that adolescents received weekly, which might reflect family income, positively predicted alcohol use. Second, past 30-day alcohol use was not significantly related to past 30-day illicit drug use. The last two findings provide more evidence that alcohol use might be subject to specific cultural influences and carry some unique risks for adolescents in Macau. These possibilities should be further explored in future research.

Despite its contributions to the research on adolescent substance use in the greater China region, this study has several major limitations that are worth noting. First, the study is based on cross-sectional surveys conducted at two different time points. Although repeated cross-sectional surveys have often been used to study the trends of individual behavior on the aggregate level, they are not as reliable as data collected from well-designed panel studies when used to assess over-time changes in behavior. Second, our analysis of factors associated with past 30-day use of cigarettes, alcohol, and illicit substances was limited to data collected in 2018. This analytic strategy was necessary because many of the factors included in the regression analysis are not available from the survey conducted in 2014. This approach, however, prevented us from examining how the roles of these predictors had changed over the two survey years. Third, the questions about illicit substances listed all controlled substances that were commonly consumed by drug users in Macau. As new psychoactive substances (NPS) continued to emerge, the list might not cover all psychoactive drugs used by the adolescents and consequently could have led to underreporting, especially on NPS. To address these limitations, future studies should consider the use of longitudinal data with more comprehensive measures of NPS. They should also incorporate measures of key factors in all waves so that researchers can examine how the roles of the predictors of adolescent substance use change over time.

## 6. Conclusions

The purpose of the study is threefold: estimating the prevalence rates of adolescent drug use, identifying the patterns of change in the behavior between 2014 and 2018, and testing the relationships between risk factors and adolescent substance use. Through the analysis of survey data collected from two representative samples of middle school and high school students in Macau, we found low-to-moderate rates of cigarette smoking and illicit drug use but a relatively high rate of alcohol use. Over the four years, cigarette smoking decreased significantly while illicit drug use remained relatively stable. Among all drug categories, those related to the new psychoactive substance (NPS) showed the largest increase. Our study identified upper-grade level students, especially the 12th graders, as the adolescent group with the highest risk for substance use. Additionally, our results indicate that living without a parent, living with both parents unemployed, a greater amount of pocket money, approval of substance use, low perception of risk from substance use, peer substance use, school problems, and availability of drugs all predicted one or more forms of substance use among adolescents in Macau. Furthermore, cigarette smoking operated as a predictor of illicit drug use while alcohol use did not appear to play the same role.

## Figures and Tables

**Figure 1 ijerph-19-07988-f001:**
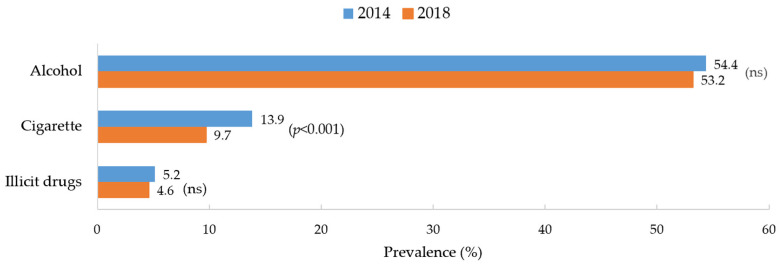
Changes in the rates of lifetime use of alcohol, cigarettes, and illicit drugs from 2014 to 2018. Note. Only significant *p* values are shown, with ‘ns’ meaning *p* values are not significant. The level of significance was set at *p* < 0.05.

**Figure 2 ijerph-19-07988-f002:**
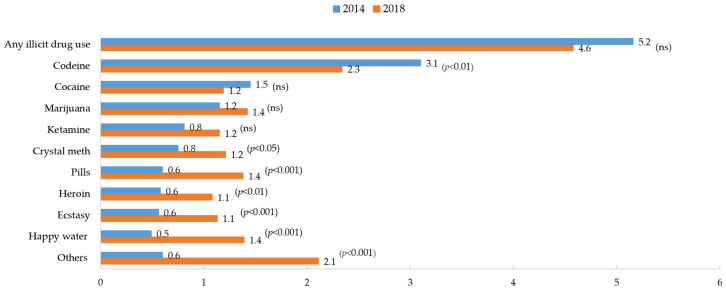
Trends in prevalence of lifetime use of 10 nominated illicit drugs from 2014 to 2018. Note. Only significant *p* values are shown, with ‘ns’ meaning *p* values are not significant. The level of significance was set at *p* < 0.05.

**Figure 3 ijerph-19-07988-f003:**
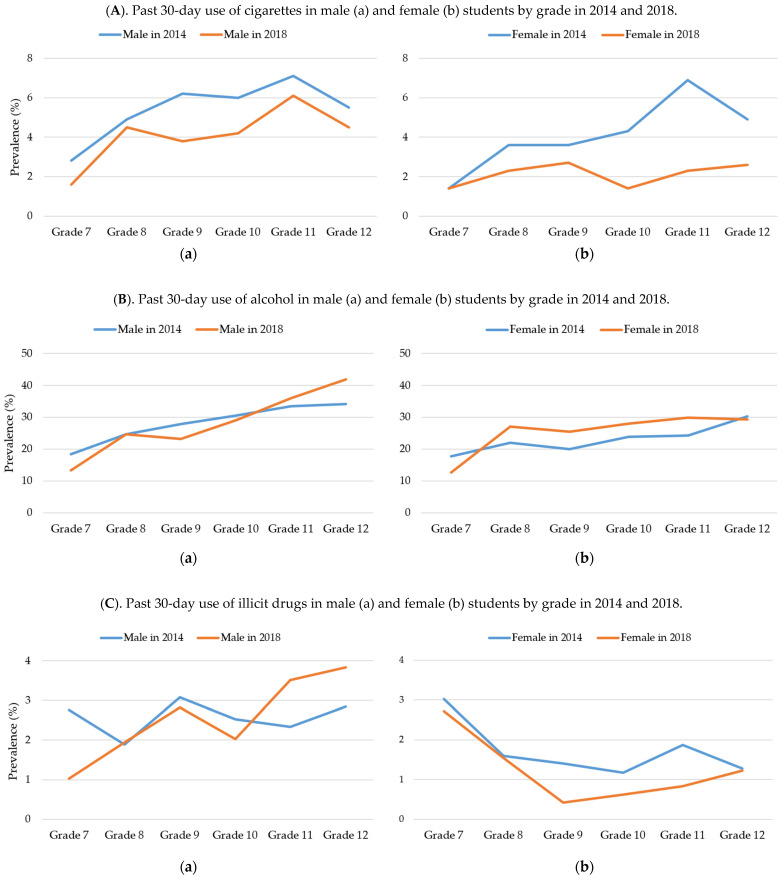
Substance use and misuse in male (**a**) and female (**b**) students by grade in 2014 and 2018.

**Table 1 ijerph-19-07988-t001:** Prevalence of past 30-day substance use among students from grade 7 to grade 12 in Macau in 2014 and 2018.

	2014 (*n* = 6782)							2018 (*n* = 6459)						
	Total	Cigarette Use		Alcohol Use		Illicit Drug Use		Total	Cigarette Use		Alcohol Use		Illicit Drug Use	
	*n* (%)	*n* (%)	*p*	*n* (%)	*p*	*n* (%)	*p*	*n* (%)	*n* (%)	*p*	*n* (%)	*p*	*n* (%)	*p*
**Total**	6782 (100)	329 (4.9)		1765 (26.0)		148 (2.2)		6459 (100)	202 (3.1)		1682 (26.0)		123 (1.9)	
**Gender**														
Male	3577 (52.7)	19 (5.5)	0.008	1021 (28.5)	<0.001	92 (2.6)	0.014	3467 (53.7)	139 (4.0)	<0.001	926 (26.7)	0.187	83 (2.4)	0.001
Female	3182 (46.9)	131 (4.1)		737 (23.2)		54 (1.7)		2973 (46.0)	63 (2.1)		751 (25.3)		37 (1.2)	
**Place of birth**														
Macau	5231 (77.1)	255 (4.9)	0.007	1397 (26.7)	0.008	106 (2.0)	<0.001	4943 (76.5)	148 (3.0)	<0.001	1297 (26.2)	0.199	89 (1.8)	<0.001
Mainland China	1063 (15.7)	37 (3.5)		235 (22.1)		18 (1.7)		1154 (17.9)	34 (2.9)		283 (24.5)		22 (1.9)	
Hong Kong	126 (1.9)	9 (7.1)		40 (31.7)		5 (4.0)		173 (2.7)	4 (2.3)		43 (24.9)		1 (0.6)	
Others	331 (4.9)	26 (7.9)		88 (26.6)		18 (5.4)		176 (2.7)	16 (9.1)		56 (31.8)		11 (6.3)	
**Grade**														
Grade 7	1045 (15.4)	22 (2.1)	<0.001	190 (18.2)	<0.001	31 (3.0)	0.495	1198 (18.5)	18 (1.5)	0.008	155 (12.9)	<0.001	21 (1.8)	0.538
Grade 8	1034 (15.2)	44 (4.3)		241 (23.3)		18 (1.7)		1141 (17.7)	40 (3.5)		293 (25.7)		20 (1.8)	
Grade 9	1086 (16.0)	54 (5.0)		263 (24.2)		25 (2.3)		1086 (16.8)	36 (3.3)		262 (24.1)		21 (1.9)	
Grade 10	1235 (18.2)	65 (5.3)		338 (27.4)		24 (1.9)		1128 (17.5)	34 (3.0)		324 (28.7)		16 (1.4)	
Grade 11	1181 (17.4)	81 (6.9)		347 (29.4)		25 (2.1)		993 (15.4)	42 (4.2)		328 (33.0)		22 (2.2)	
Grade 12	1179 (17.4)	62 (5.3)		381 (32.3)		25 (2.1)		913 (14.1)	32 (3.5)		320 (35.0)		23 (2.5)	

Note: Bolded content are the demographic information controlled for in the study’s analysis.

**Table 2 ijerph-19-07988-t002:** Sample characteristics in 2018 (*n* = 6459).

	*n* (%)
Total	6459 (100)
**Monthly pocket money**	
0 MOP	976 (15.1)
1–500 MOP	2981 (46.2)
501–1000 MOP	924 (14.3)
More than 1000 MOP	1578 (24.4)
**Family structure**	
Both parents are alive	6170 (95.5)
Only father is alive	69 (1.1)
Only mother is alive	177 (2.7)
Both parents are deceased	22 (0.3)
**Parental employment**	
Both parents are employed	4706 (72.9)
Only father is employed	1093 (16.9)
Only mother is employed	475 (7.4)
Both parents are unemployed	147 (2.3)
**School problems**	
No	5704 (88.3)
Yes	755 (11.7)
**Perception of risk of** **cigarettes smoking**	
Unknown/none/slight/moderate risk	2841 (44.0)
Great risk	3618 (56.0)
**Perception of risk of alcohol use**	
Unknown/none/slight/moderate risk	3111 (48.2)
Great risk	3348 (51.8)
**Perception of risk of illicit drug use**	
Unknown/none/slight/moderate risk	824 (12.8)
Great risk	5635 (87.2)
**Approval of cigarettes smoking**	
Unknown/somewhat/strongly disapprove	5408 (83.7)
Somewhat/strongly approve	1051 (16.3)
**Approval of alcohol use**	
Unknown/somewhat/strongly disapprove	4931 (76.3)
Somewhat/strongly approve	1528 (23.7)
**Approval of illicit drug use**	
Unknown/somewhat/strongly disapprove	6195 (95.9)
Somewhat/strongly approve	264 (4.1)
**Perceived access to illicit drugs**	
Unknown/fairly/very difficult or probably impossible	5071 (78.5)
Fairly/very easy	1388 (21.5)
**Peers’ cigarette smoking**	
None	5416 (83.9)
At least one	1043 (16.1)
**Peers’ alcohol use**	
None	4667 (72.3)
At least one	1792 (27.7)
**Peers’ illicit drug use**	
None	6350 (98.3)
At least one	109 (1.7)
**Past 30-day use of cigarette**	
No	6257 (96.9)
Yes	202 (3.1)
**Past 30-day use of alcohol**	
No	4777 (74)
Yes	1682 (26.0)
**Past 30-day use of illicit drugs**	
No	6336 (98.1)
Yes	123 (1.9)

Note: Demographic information (gender, grade, birth of place) included in the regression analysis that is already presented in Table 1 is omitted from Table 2. MOP: Macau Pataca, official currency in Macao. Bolded content are the names of the variables and factors used the study’s analysis.

**Table 3 ijerph-19-07988-t003:** Factors associated with past 30-day substance use in 2018 (*n* = 6459).

	Model 1		Model 2		Model 3	
	Cigarette Use		Alcohol Use		Illicit Drug Use	
	OR (95% CI)	*p*	OR (95% CI)	*p*	OR (95% CI)	*p*
**Birth of place**						
Macau	ref		ref		ref	
Mainland China	0.85 (0.54–1.33)	0.484	0.78 (0.65–0.92)	0.004	1.32 (0.77–2.25)	0.315
Hong Kong	0.83 (0.26–2.64)	0.751	1.21 (0.83–1.79)	0.325	0.37 (0.05–2.85)	0.342
Others	2.92 (1.43–5.96)	0.003	1.08 (0.74–1.58)	0.694	1.84 (0.75–4.54)	0.184
**Monthly pocket money**						
0 MOP	ref		ref		ref	
1–500 MOP	0.30 (0.17–0.53)	<0.001	1.26 (1.03–1.55)	0.026	0.66 (0.37–1.17)	0.152
500–1000 MOP	0.46 (0.25–0.85)	0.013	1.42 (1.11–1.81)	0.005	0.56 (0.26–1.17)	0.122
More than 1000 MOP	0.70 (0.43–1.16)	0.169	1.77 (1.43–2.20)	<0.001	0.62 (0.34–1.12)	0.110
**Family structure**						
Both parents are alive	ref		ref		ref	
Only father alive	1.54 (0.42–5.66)	0.515	1.36 (0.76–2.44)	0.296	(Empty) ^#^	
Only mother alive	1.55 (0.65–3.65)	0.321	0.72 (0.47–1.11)	0.140	0.29 (0.03–2.46)	0.258
Both parents are deceased	8.27 (2.00–34.15)	0.004	0.31 (0.08–1.24)	0.098	3.15 (0.67–14.69)	0.144
**Parents’ employment**						
Both parents are employed	ref		ref		ref	
Just father employed	0.90 (0.55–1.47)	0.675	1.01 (0.85–1.21)	0.898	0.67 (0.36–1.27)	0.218
Just mother employed	1.13 (0.60–2.12)	0.710	1.21 (0.94–1.56)	0.142	0.73 (0.28–1.88)	0.511
Both parents are unemployed	2.13 (1.01–4.48)	0.048	0.94 (0.61–1.46)	0.792	0.93 (0.31–2.77)	0.898
**Approval of cigarettes smoking**						
Disapprove	ref					
Approve	10.27 (6.98–15.11)	<0.001				
**Approval of alcohol drinking**						
Disapprove			ref			
Approve			2.87 (2.48–3.32)	<0.001		
**Approval of illicit drug using**						
Disapprove					ref	
Approve					3.54 (2.04–6.15)	<0.001
**Perception of risk for cigarette smoking**						
Not a great risk	ref					
Great risk	0.35 (0.22–0.54)	<0.001				
**Perception of risk for alcohol drinking**						
Not a great risk			ref			
Great risk			0.79 (0.69–0.90)	<0.001		
**Perception of risk of illicit drug using**						
Not a great risk					ref	
Great risk					0.43 (0.26–0.69)	<0.001
**Perceived access to illicit drugs**						
Difficult					ref	
Easy					2.16 (1.41–3.31)	<0.001
**Peer’s** **cigarette smoking**						
None	ref		ref		ref	
At least one	3.35 (1.57–7.17)	0.002	1.12 (0.90–1.39)	0.320	1.27 (0.55–2.92)	0.573
**Peer’s alcohol drinking**						
None	ref		ref		ref	
At least one	0.28 (0.13–0.60)	0.001	1.53 (1.28–1.83)	<0.001	0.71 (0.34–1.51)	0.376
**Peer’s illicit drug use**						
None	ref		ref		ref	
At least one	1.38 (0.64–2.96)	0.408	1.18 (0.71–1.96)	0.518	6.27 (2.92–13.45)	<0.001
**Past 30-day cigarette use**						
No			ref		ref	
Yes			2.40 (1.62–3.57)	<0.001	5.60 (3.13–10.01)	<0.001
**Past 30-day alcohol use**						
No	ref				ref	
Yes	2.89 (1.91–4.37)	<0.001			1.52 (0.92–2.50)	0.102
**Past 12-month drunk**						
No	ref		ref		ref	
Yes	5.42 (3.69–7.99)	<0.001	8.72 (7.00–10.87)	<0.001	1.38 (0.77–2.48)	0.276
**School Problems**						
No	ref		ref		ref	
Yes	1.91 (1.20–3.04)	0.007	1.29 (1.05–1.57)	0.014	1.37 (0.77–2.43)	0.290

Note: Two demographic variables (gender, grade) were controlled in the final multivariable regression models. MOP: Macau Pataca, official currency in Macao. ^#^. No observed cases. Bolded content are the names of the variables and factors used the study’s analysis.

## Data Availability

The data presented in this study are available on request from the first author.

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
