# Peer review of "The Changing Patterns and Correlates of Adolescent Substance Use in China’s Special Administrative Region of Macau"

_ijerph, 2022, doi:10.3390/ijerph19137988_

Round 1

Reviewer 1 Report

The field of investigation of this paper is interesting and although the data are referred to 2014 and 2018, it be considered actual. The tables are quite exhaustive but I think that reporting the data about the sample configuration (males, females, Mage etc) also in the text is more adequat.

Moreover is necessary to improve Measures with Alfa Cronbach coefficient. This is important as you have modify some scales.

Reviewer 2 Report

Adolescence is recognized as the period for onset of behaviors and conditions that not only affect health limited to that time but also lead to adulthood disorders. Unhealthy behaviors such as smoking, drinking, and illicit drug use often begin during adolescence; they are closely related to increased morbidity and mortality and represent major public health challenges. Unemployment, poor health, accidents, suicide, mental illness, and decreased life expectancy all have drug misuse as a major common contributing factor. Substance abuse has a major impact on individuals, families, and communities as its effects are cumulative, contributing to costly social, physical, and mental health problems. In presented manuscript the Authors used data collected from secondary school students in 2014 and 2018 to provide population-based estimates of the prevalence rates of lifetime and past 30-day substance use among Macau. Therefore, current study is timely and scientifically sound. In my opinion:

·       The abstract is structured correctly, it contains a summary of all important informations from the manuscript.

·       Key words have been appropriately selected.

·       The introduction is an interesting and comprehensive preludy to the study, and appropriately introduces the presented topic.

·       • The methods are accurately described.

·       • The results are presented in a clear manner and supported by numerous figures and tables, which make them easier to understand.

·       The discussion contains a detailed description of the obtained results.

·       Literature was correctly selected.

Therefore, I only have a few reservations. Major:

In my opinion, part of the Literature review is unnecessary. As it stands, it only extends the article. Rather, it should be presented and published by the Authors as a mini review, in a separate manuscript. In addition, my doubts are raised by the validity of the analysis carried out by Authors, because in the literature review part there are also data for the Macau population in the years 2014 or 2018, which is the subject of the presented manuscript. Therefore, it seems that there is a lack of novelty in presented manuscript.

Minor:

The entire manuscript should be written in one font. This needs improvement.

The section conclusions i stoo long it should contain only the most important conclusions.

There is no limitations section, this should be completed.
